# Prediction of Parkinson’s Disease Using Machine Learning Methods

**DOI:** 10.3390/biom13121761

**Published:** 2023-12-08

**Authors:** Jiayu Zhang, Wenchao Zhou, Hongmei Yu, Tong Wang, Xiaqiong Wang, Long Liu, Yalu Wen

**Affiliations:** 1Department of Health Statistics, School of Public Health, Shanxi Medical University, No. 56 Xinjian South Road, Yingze District, Taiyuan 030001, China; zjiayu0217@gmail.com (J.Z.); zhouwenchao157@gmail.com (W.Z.); yu@sxmu.edu.cn (H.Y.); tongwang@sxmu.edu.cn (T.W.); 2Department of Epidemiology and Biostatistics, Southeast University, 87 Ding Jiaqiao Road, Nanjing 210009, China; xiaqiong.wang@auckland.ac.nz; 3Department of Statistics, University of Auckland, 38 Princes Street, Auckland Central, Auckland 1010, New Zealand

**Keywords:** machine learning, Parkinson’s disease, polygenic risk scores, risk prediction model, SHAP value

## Abstract

The detection of Parkinson’s disease (PD) in its early stages is of great importance for its treatment and management, but consensus is lacking on what information is necessary and what models should be used to best predict PD risk. In our study, we first grouped PD-associated factors based on their cost and accessibility, and then gradually incorporated them into risk predictions, which were built using eight commonly used machine learning models to allow for comprehensive assessment. Finally, the Shapley Additive Explanations (SHAP) method was used to investigate the contributions of each factor. We found that models built with demographic variables, hospital admission examinations, clinical assessment, and polygenic risk score achieved the best prediction performance, and the inclusion of invasive biomarkers could not further enhance its accuracy. Among the eight machine learning models considered, penalized logistic regression and XGBoost were the most accurate algorithms for assessing PD risk, with penalized logistic regression achieving an area under the curve of 0.94 and a Brier score of 0.08. Olfactory function and polygenic risk scores were the most important predictors for PD risk. Our research has offered a practical framework for PD risk assessment, where necessary information and efficient machine learning tools were highlighted.

## 1. Introduction

Parkinson’s disease (PD) is currently the fastest-growing neurodegenerative disease in the world [1], with its prevalence and socioeconomic burden second only to Alzheimer’s disease [2]. It is primarily characterized by symptoms such as muscle stiffness, slow movement, and resting tremors [3]. It typically occurs in the middle-aged and elderly population, and its prevalence is gradually increasing as the population ages. It is estimated that by 2040, the number of PD patients worldwide will reach 12 million [4]. The exact causes of PD are still unknown, and the symptoms of PD appear relatively late in the pathological process, by which time the diagnosis of PD has already been delayed [5]. Therefore, accurately predicting PD in its early stages is crucial for promoting health and effective treatments.

PD, as a complex and heterogeneous multifactorial disease, is influenced by both genetic and environmental factors. Demographic characteristics (e.g., age, gender, education, race/ethnicity, and family history) have already been shown to have predictive power in PD [6]. For example, compared with females, the risk of PD for males is increased by 3.27 times [7]. Individuals with first or second-degree relatives having PD have a 4.2-fold increased risk of PD [8]. In addition, studies have shown that the prevalence of left-handedness is higher among PD patients, suggesting a connection between hand dominance and the side of asymmetric disease manifestation [9]. Some of the routinely collected measures from hospital admission have shown to be associated with PD. For example, height is positively correlated with the risk of late-onset PD in males, while obesity indicated by body mass index is associated with a higher mortality rate in male PD patients [10]. Timely treatment of orthostatic hypotension and supine hypertension can improve the quality of life of Parkinson’s patients and prevent the occurrence of short-term and long-term complications [11]. Clinical assessments, which evaluate olfactory function, cognitive function, sleep quality, emotional symptoms, autonomic dysfunction, and language fluency, are often used as tools for PD diagnosis and treatments. For example, the olfactory function, typically assessed through the University of Pennsylvania Smell Inventory Test (UPSIT), has the potential for early disease diagnosis [12]. The Montreal Cognitive Assessment (MoCA) that assesses the risk of cognitive decline in PD can screen for dementia in PD patients. Table 1 summarizes the commonly used tests and questionnaires for the clinical assessment of PD. Biomarkers, such as cerebrospinal fluid (CSF) biomarkers and uric acid, have provided additional information for PD risk. For example, CSF biomarkers including alpha-synuclein (α-Syn), amyloid β1-42 (Aβ_42_), total tau (t-tau), and phosphorylated tau (p-tau) are associated with cognitive decline and can serve as independent predictors of cognitive impairment in PD [13]. Low levels of uric acid are significantly correlated with motor impairments [14]. Genetic factors have also played a significant role in PD, although less than 1% of cases are caused by a single gene mutation [2]. Indeed, research indicates that the vast majority is sporadic, showcasing complex and polygenic genetic influences of PD [15]. Genome-wide association study (GWAS) analysis has made substantial contributions in supporting the genetic aspect of PD etiology, and Blauwendraat et al. estimated that the heritability of PD is around 27% [16]. The latest, largest GWAS detected 90 independent PD-associated loci, which can explain 16% to 36% of the genetic risk [17]. Polygenic risk scores (PRS) that are calculated as the weighted sum of genetic variants are commonly used tools to quantify genetic susceptibility to a specific trait or disease [18], and recent studies have shown that PRS that are constructed using genome-wide data have better prediction power than those built with significant disease-associated loci. For example, our latest study has obtained an AUC of 0.75 using genome-wide data [19], whereas the model constructed by Nalls et al. that is built with known PD-related loci has only achieved an AUC of 0.64 [17]. While demographic variables, hospital admission examinations, clinical assessments, biomarkers, and PRS all provide important information for PD risk, it remains unclear how to best integrate them to facilitate the early detection of PD.

Machine learning (ML) including deep learning techniques have been widely used in medical studies [20], including the investigation of PD [21,22]. For example, Shahid et al. used a deep neural network model to predict the progression of PD [21]. Harvey et al. employed random forest, elastic net, support vector machine, and conditional inference forest to predict cognitive outcomes in PD, where the impacts of clinical and biofluid predictors are modeled [23]. Makarious et al. integrated transcriptomics, genetics, and clinical data into GenoML, an automated ML open-source Python package that develops a peri-diagnostic model to predict PD risk [24]. Park et al. incorporated variables such as anthropometric data, laboratory data, lifestyle factors, and others into neural networks, gradient boosting machines, and random forest algorithms, achieving a maximum AUC of 0.779 [25]. Existing ML techniques can range from simple models (e.g., decision tree and logistic regression) to much more complex models (e.g., deep neural network) and their prediction performance can differ dramatically depending on the diseases of interest and their associated predictors. Although these ML models help to explain PD risks, it can be hard to determine the appropriate ML models and their associated predictors when assessing PD risk, as existing PD prediction models rely on different ML algorithms with quite different sets of predictors. There are few systematic comparisons among the existing ML algorithms for PD risk prediction, where each ML algorithm is constructed with the same amount of input information. There also lack investigations on the essential information for accurate PD risk assessment for each ML method, where the cost and accessibility of each predictor have been considered. Indeed, some of the PD risk factors can be invasive (e.g., CSF biomarkers) and expensive (e.g., PET imaging), which can pose challenges for their collection. Therefore, it is of great practical importance to evaluate the prediction power of different ML models on PD risk assessment and determine the essential information needed for each ML model.

To better facilitate the clinical evaluation of PD risk, our study aims to identify the most appropriate ML algorithms as well as the suitable set of predictors while taking the cost and accessibility of each predictor into account. Using data obtained from the Parkinson’s Progression Markers Initiative (PPMI), we first grouped predictors based on their cost and accessibility and evaluated their relative contributions to PD risks. We then systematically evaluated the PD risk prediction accuracy for the commonly used ML algorithms that range from the most classic methods to the latest developments. Finally, we explored explanations for our best predictive models.

## 2. Materials and Methods

### 2.1. Baseline Data

PPMI (https://www.ppmi-info.org, accessed on 20 August 2023) [26] is an international, multicenter, prospective cohort study aimed at discovering and validating new biomarkers for the progression of PD. PPMI employs advanced imaging techniques and clinical monitoring to comprehensively assess PD patients, prodromal individuals, and healthy controls. Its primary objective is to identify biomarkers of disease progression, with the ultimate goal of expediting therapeutic trials to reduce the progression of disability in PD. This comprehensive dataset provides valuable insights into various aspects of PD and is instrumental for research and analysis in the field. The original study obtained approval from the Institutional Review Boards at each trial site institution, and participants were provided written informed consent.

The PPMI dataset was downloaded on 20 August 2023. We considered thirty-five PD-related variables and categorized them into four groups based on their accessibility, including demographic variables (S1 in Table 2), hospital admission examinations (S2 in Table 2), clinical assessments (S3 in Table 2), and biomarkers (S4 in Table 2). Participants with excessive missing data (>15% for each variable) were excluded. A total of 564 PD patients and 183 controls were included in our final analyses. For the remaining missing data, we followed the same procedure as Harvey et al. [21] and imputed the continuous and categorical variables using the median and mode, respectively.

### 2.2. Genomic Data, Quality Control, and PRS Construction

We downloaded the whole-genome sequencing data (Project 118; July 2018 version) from the Laboratory of NeuroImaging Image Data Archive PPMI project website (https://ida.loni.usc.edu/, accessed on 20 August 2023). DNA samples were prepared using the Illumina HiSeq X Ten Sequencer (Illumina, San Diego, CA, USA). In our study, autosomal single nucleotide polymorphisms (SNPs) were extracted, and we used the UCSC genome browser’s liftover tool to convert genome coordinates from hg38 to hg19. We excluded individuals with a missing rate larger than 10%, and a total of 747 participants were retained. For the genetic data, the following exclusion criteria were applied: (1) genetic variants with missingness > 0.1; (2) genetic variants with a minor allele frequency (MAF) < 5%; (3) *p*-value < 1 × 10^−10^ in cases and < 1 × 10^−6^ in controls for Hardy–Weinberg equilibrium [27]; and (4) unclear, mismatched, and duplicate SNPs. For the remaining SNPs, we further performed imputation using PLINK v1.9 with its default settings.

The summary statistics for 7.8 million SNPs, one of the largest for PD to date, included samples from Nalls et al. [28], 23andMe Web-Based PD study [29], International Parkinson’s Disease Genomics Consortium, and 13 additional datasets [17]. They are derived based on a total of 37,700 PD cases, 18,600 proxy-cases, and 1.4 million controls. After excluding SNPs with a MAF less than 0.01 and further matching with the PPMI genetic data, 5,466,717 SNPs were kept for the PRS constructions.

We built PD-PRS using the SDPR method, a flexible and robust Bayesian linear mixed model that can accommodate complex diseases and efficiently model effect sizes of various distributions [30]. We did not consider the other methods (e.g., clumping and thresholding (C+T) [31] and LDpred [32]) mainly because the robustness and predictive power of PRS depend on whether the modeling assumptions align with the underlying disease models and SDPR has shown to have better predictive power across a range of complex diseases [30], including PD [19]. We have provided the technical details of the SDPR method in the Appendix A. In our analyses, we set the number of Markov Chain Monte Carlo iterations for SDPR to be 500 and the other parameters were kept at their default settings.

### 2.3. The Prediction of PD Risk

The goal of our study is to identify the most appropriate ML algorithms as well as the suitable set of predictors that can accurately predict the risk of PD, while taking into account the accessibility and cost of obtaining the information on the relevant variables. Figure 1 provides an overview of the research. We considered eight widely used ML methods to predict PD risk. These methods include decision tree, K nearest neighbors, naïve Bayes, neural network, penalized logistic regression, random forest, support vector machine (SVM), and Extreme Gradient Boosting (XGBoost). Appendix A summarizes the corresponding settings for each method.

To take the cost and accessibility into consideration, we designed five modeling strategies for each ML method (Figure 1), where the difficulties of obtaining the relevant predictors are increasing either due to the cost or the procedure itself. We started by considering the scenario where only routinely collected demographic variables are modeled (i.e., scenario I). We then gradually incorporated hospital admission examinations (scenario II), PD-relevant clinical assessments that are relatively easy to obtain (scenario III), polygenic risk scores that are built from relatively expensive genotype data (scenario IV), and invasive biomarkers (scenario V) into the model built from the previous scenario. Scenario V has the maximum number of variables, and it modeled the impact from all 35 PD-relevant variables and the PRS.

For each ML algorithm and modeling strategy, we utilized the idea of cross-validation. We randomly sampled 70% of the data to train the model and used the remaining 30% to validate the results. To reduce the chance finding, we repeated this process 20 times. We assessed the classification accuracy using the testing set, where the true positive rate (TPR), true negative rate (TNR), area under the curve (AUC), Brier score, Matthews correlation coefficient (MCC), and accuracy [33] are reported.

Finally, we used SHAP values to provide an intuitive explanation for the selected variables, with SHAP values calculated as the mean based on results from 20 random samples. SHAP analysis, proposed by Lundberg and Lee, is frequently utilized for interpreting black-box machine learning models [34]. This method evaluates the contribution of each feature to model predictions by considering their impact across various combinations, utilizing Shapley values from coalitional game theory, transforming the feature space of predictive models into a clinical variable space to quantify their impact on the model output. In our research, we used kernel SHAP that employs weighted local linear regression to estimate SHAP values and visualizes the results. We utilized the SHAP variable importance plot to demonstrate the influence of each modeled variables on predictions and employed the SHAP summary plot, a beeswarm plot, to obtain information regarding the magnitude and directionality of contributions from each variable in predicting PD. SHAP analysis was conducted using the Python SHAP library (0.40.0).

## 3. Results

A total of 564 patients and 183 controls were included in our study. Baseline characteristics, including demographic variables, hospital admission examinations, clinical assessments, and biomarkers are summarized in Table 2.

### 3.1. The Impact of Different Modeling Strategies

Figure 2 presents the prediction performance derived from the cross-validation testing samples across five different scenarios, where we progressively increased the number of predictors based on their ease of acquisition. We observed that, as the groups of predictors increased, the predictive performance of all models generally improved, especially for scenarios III, IV, and V. Comparing scenarios I and II, we found that the further inclusion of variables obtained at hospital admission examinations into the model that was built with demographic variables did not significantly improve the predictive performance (*p*-value > 0.05 for the comparison between AUCs based on the Delong test [35]). Even though some of the hospital admission variables (e.g., height, weight, and blood pressure) have shown to be PD-related [10], they did not offer additional prediction power across all the ML models considered. Indeed, we noticed that for some ML models (i.e., decision tree, naïve Bayes, and random forest), the inclusion of these variables could result in a significant worse performance with regards to accuracy and Brier score.

It is worth noting that the inclusion of clinical assessments into the risk prediction models significantly boosted the prediction accuracy regardless of the ML methods. As expected, incorporating clinical assessments (scenario III) into the predictive models built from scenario II can significantly improve the prediction accuracy for all ML models, especially for SVM with an increase of 54.50%. Under scenario III, both penalized logistic regression and XGBoost can achieve AUCs of above 0.9. The addition of PRS (scenario IV) into the model that has demographic variables, hospital admission examinations, and clinical assessments (i.e., scenario III) also resulted in an improvement in the prediction performance for PD. We noticed that in scenario IV, the AUC for penalized logistic regression and XGBoost could exceed 0.92, which is significantly higher than that from scenario III (*p*-value < 0.05 based on the Delong test). Additionally, the inclusion of invasive biomarkers into the model built with demographic variables, admission examinations, clinical assessments, and PRS (i.e., scenario IV) further improved predictive performance. In comparison to scenario IV and scenario V, the AUC values for penalized logistic regression, random forest, and naïve Bayes increased by only 0.23%, 0.27%, and 0.38%, respectively. However, the AUC for the other methods did not improve and, in some cases, even showed a slight decrease. Other measures also exhibited a similar trend that was similar to AUC.

In summary, the modeling strategy (i.e., scenario IV) that included demographic variables, hospital admission examinations, clinical assessments, and PRS represented the best modeling approach for PD. On one hand, the AUC, accuracy, Brier score, and MCC in scenario IV are all not significantly different from scenario V that has invasive biomarkers further incorporated (*p*-values > 0.05 based on the Mann–Whitney U and Delong tests), indicating that these biomarkers cannot offer additional power for PD risk prediction. On the other hand, the AUC, accuracy, Brier score, and MCC in scenario IV are all significantly better than scenario III where PRS is excluded (*p*-value < 0.05), suggesting that genetic factors help to further improve prediction accuracy. Therefore, we believe that scenario IV, which combines demographic variables, admission examinations, clinical assessment scales, and PRS, represents the best model for predicting PD, yielding superior results in terms of AUC and all other metrics without requesting invasive and time-consuming measurements.

### 3.2. The Impact of Different Machine Learning Methods

We mainly focused on the comparisons among different ML methods under the best performed scenario that included demographic variables, hospital admission examinations, clinical assessments, and PRS in the model (i.e., scenario IV). Figure 3 shows the accuracy, AUC, Brier score, and MCC among the eight ML methods under this scenario. As expected, different ML methods can have very different predictive powers. Using accuracy as the criterion, penalized logistic regression, random forest, XGBoost, and SVM perform the best, followed by decision tree and neural network, with K nearest neighbors and naïve Bayes performing the worst. Using AUC as the criterion, penalized logistic regression and XGBoost are among the top-tier, followed by neural network, SVM, naïve Bayes, random forest, with decision tree, and K nearest neighbors being the worst. For Brier score, penalized logistic regression and XGBoost are among the best, followed by random forest, SVM, and neural network, with decision tree, K nearest neighbors, and naïve Bayes being the worst. Similar trends hold for MCC, except that penalized logistic regression, random forest, SVM, and XGBoost perform similarly. Taking all four criteria together, the PD risk prediction models built with penalized logistic regression and XGBoost are the most accurate and robust, and the AUCs for penalized logistic regression and XGBoost achieved 0.94 and 0.92, respectively.

Appendix A summarizes the prediction performance for the ML methods under other scenarios (i.e., scenarios I to III and scenario V), and Table 3 provides the corresponding average prediction performance across all the scenarios. As we progressively moved from scenario I to scenario V, the prediction performance for all machine learning methods generally increased. We noticed that models incorporating embedded variable selection techniques generally exhibit better feature relevance and model explainability. This leads to an improvement in model performance since the model can focus on the most relevant variables, reducing noise and the risk of overfitting. On the other hand, models without embedded variable selection techniques often have a larger feature space, encompassing more variables. While this approach can capture a broader range of information, it frequently results in increased complexity, potentially leading to overfitting and a decrease in model explainability. Consequently, these models sometimes struggle to generalize well to new data or make accurate predictions. Notably, under all the scenarios considered, the simple penalized logistic regression has consistently showed the best or close to the best results. While tree-based methods and neural networks are particularly powerful in capturing non-linear effects, our study suggests that for PD risk prediction, penalized logistic regression remains the most effective approach. This may be because it can effectively handle feature selection and classification tasks specific to the problem while maintaining model simplicity.

### 3.3. Explainability and the Importance of Predictors

Regardless of the ML methods, the models that were built with demographic variables, hospital admission examinations, clinical assessments, and PRS exhibited the best predictive performance for PD. Therefore, we focused on the explainability of models constructed with all predictive factors from scenario IV. Figure 4 displays the top 10 factors in descending order of their SHAP values that reflect the impact on prediction performance. The importance of each predictor is generally consistent across different ML methods, with UPSIT consistently ranked the first and PRS ranked among the top three. This is consistent with the findings from Figure 2, where we gradually increased the predictors according to their accessibility. Incorporating clinical assessments that include UPSIT into prediction models significantly improves the prediction accuracy (scenario III vs. II) and further adding PRS into scenario IV also enhances the prediction performance. In addition to UPSIT and PRS, MoCA, the Categorical REM Sleep Behavior Disorder Questionnaire (RBDSQ), and Assessment of Autonomic Dysfunction (SCOPA) are among the top 10 important predictive factors for all models. Family history, State Trait Anxiety Test, age, blood pressure, and others also hold significant predictive value. Furthermore, the SHAP summary plot (Figure 5) further summarizes the direction of the relationships based on the results of a single random sampling. We focused on examining the SHAP analysis results for penalized logistic regression and XGBoost. We found that these two methods have a high degree of overlap in SHAP analysis. PRS, RBDSQ, SCOPA, family history, and heart rate have high SHAP values and are positively associated with higher PD risk, while lower values of UPSIT, MoCA, and systolic blood pressure increase the risk of PD.

## 4. Discussion

PD is a complex neurological disorder with an as yet incompletely understood etiology, and it currently lacks effective treatments [36]. Studies have indicated that identifying PD in its prodromal stage and implementing protective interventions can be highly beneficial for PD patients [37]. Therefore, the development of an accurate and cost-effective model to aid in the early diagnosis of PD is of paramount importance. In our study, we aimed at constructing PD risk prediction models with cost-effective and non-penetrance measures. We systematically investigated the prediction performance of eight ML methods, including decision tree, K nearest neighbors, naïve Bayes, neural network, penalized logistic regression, random forest, SVM, and XGBoost, under a variety of modeling strategies. We found that prediction models built with demographic characteristics, hospital admission examinations, clinical assessments, and genetic data can achieve similar prediction performance as those with invasive CSF biomarkers further incorporated. We also found that penalized logistic regression and XGBoost are the most accurate PD risk prediction methods, with AUCs reaching 0.94 and 0.92, respectively. This indicates that PD risk can be accurately assessed with cost-effective and practical measures. Our SHAP analysis further suggests that UPSIT and PRS are among the top three important predictors for PD. The effective tool UPSIT for detecting olfactory dysfunction in PD and the PRS that aggregates a large number of genetic variations to assess the genetic risk of PD are of significant importance for the early identification of PD.

Our research has found that integrating multidimensional data significantly improves the performance of PD prediction. In our study, we have grouped predictors based on their cost and accessibility. Although demographic characteristics and general hospital admission measures are associated with PD [38], they are not particularly powerful for prediction (Figure 2). As expected, PD-related clinical assessments are the most relevant for PD risk predictors. As shown in Figure 4, UPSIT, which measures olfactory dysfunction, offers the most valuable information for PD prediction, and this is consistent with the existing studies [7]. UPSIT, as a non-invasive olfactory testing tool, indicates a higher risk of PD when the score is lower, reflecting the presence of olfactory problems in individuals. Studies have shown that genetic factors contribute to PD risk with an estimated heritability of 27% [16]. Our study is consistent with the findings of Makarious et al. [24]. By further adding genetic factors into the prediction models built with demographic variables, hospital admission examinations, and clinical assessments, the AUC significantly improved (e.g., for penalized logistic regression, increased by 2.15%). Moreover, compared to scenario III, the model’s performance was substantially enhanced with the addition of PRS, indicating that ignoring the contributions of genetic factors can lead to reduced prediction performance. For example, Chairta et al. combined a PRS composed of twelve SNPs with seven environmental factors to predict PD, emphasizing the potential of using genetic factors for predicting PD risk [39]. It is worth noting that with PRS incorporated (i.e., scenario IV), the prediction models built by penalized logistic regression and XGBoost methods can be quite accurate with penalized logistic regression having an AUC of 0.94 and Brier score of 0.08. Interestingly, we noticed that further adding invasive CSF biomarkers into this prediction model did not significantly improve prediction performance (scenario V vs. IV). Although CSF biomarkers, such as α-Syn, Aβ_42_, t-tau, and p-tau, are demonstrated to be informative for PD risk [40], they generally do not offer additional predictive power when sufficient information (i.e., clinical assessments and PRS) are already included in the model. Indeed, the prediction models built from this study exceed those where invasive or expensive measures are included. For example, Lewitt et al. incorporated compounds from the CSF into an SVM to predict PD, but achieved only an AUC of 0.79 [41]. Trezzi et al. collected CSF through lumbar puncture and applied a non-targeted metabolomics approach based on gas chromatography coupled to mass spectrometry to analyze the metabolic changes in PD patients, achieving an AUC of 0.833 [42]. The use of expensive and invasive measure can limit the practical applications of the models [43], especially for PD early diagnosis. Our study offers a more convenient and cost-effective means for evaluating PD risk without sacrificing the robustness and accuracy of the assessment.

Machine learning techniques, including deep learning models, have advanced rapidly over the past decades, and they have greatly facilitated disease risk prediction and decision making in medicine [44]. However, there is no ML method that works universally across a range of diseases, and it remains unclear which ML method suits PD risk prediction best. While simple models (e.g., logistic regression) are usually easy to implement and interpret, they are not particularly powerful in capturing complex relationships. On the contrary, complex models (e.g., neural network) are designed for modeling predictive effects of various types, but their robustness as well as explainability are usually compromised. In our study, we systematically compared the prediction performance of eight widely used ML methods for PD risk prediction. We found that the simple penalized logistic regression exhibited the best or close to the best prediction performance. This is consistent with previous studies, which indicated that more complex algorithms may not necessarily outperform logistic regression in clinical practice [45]. We also found that XGBoost, based on gradient boosting decision trees and employing second-order Taylor expansion for loss function computation, exhibited the second-best prediction performance. Surprisingly, unlike the field of natural language process and imaging recognition where neural network models have achieved state-of-the-art performance [46], the deep neural network models usually perform worse than the classic ML methods (e.g., SVM) for PD risk prediction. This could be attributed to the limited sample size in our study or the over-parameterized nature of neural networks that results in overfitting. Indeed, the determination of the best network architecture and the corresponding parameters (e.g., the batch size, the number of hidden layers, and learning rate) can be a challenging task [47]. It is worth emphasizing that in the context of medical diagnosis or disease prediction, reducing the number of false negatives can be crucial, as missing potential cases can lead to delayed intervention and treatments. For the modeling strategy IV that we recommend, the penalized logistic regression model has a false positive rate (FPR) and false negative rate (FNR) of 26.667% and 6.296%, respectively, with a cutoff value determined using the highest Youden Index being 0.726. On the other hand, the XGBoost model has a FPR and FNR of 27.136% and 5.652%, respectively, with a cutoff value of 0.896 (Table 3). This indicates that the XGBoost model misses fewer true positives. However, the penalized logistic regression model is more effective at preventing false positives. As shown in Appendix A, we have plotted the ROC curves for penalized logistic regression and XGBoost based on the best results from 20 random samples. Early treatment can significantly improve the quality of life for patients, so reducing the FNR could be more important. Under such a scenario, the XGBoost method could be more suitable than the penalized logistic regression.

In this study, variable importance analysis, as shown in Figure 4, indicates that UPSIT, PRS, MoCA, RBDSQ, and SCOPA are the most critical variables influencing the model’s prediction of PD. Following them in importance are family history, State-Trait Anxiety Test, age, and blood pressure. Among the demographic variables, age and family history are also significant factors contributing to the occurrence of PD. Family history has consistently been regarded as an independent risk factor for PD [48]. In the hospital admission examinations, supine heart rate is a risk factor for PD, while systolic blood pressure is a protective factor. A recent study indicated that with the progression of the disease, PD patients experience a gradual decrease in systolic blood pressure [38]. A cohort study suggested that men with a resting heart rate > 100 beats per minute have a 1.47 times higher risk of developing PD compared to men with a heart rate of 60–100 beats per minute [49], consistent with our findings. In clinical assessments, higher RBDSQ and SCOPA scores, and lower UPSIT and MoCA scores were associated with a higher PD risk, aligning with the fact that sleep behavior disorders, autonomic dysfunction, olfactory dysfunction, and cognitive decline can increase the risk of developing PD [23]. Furthermore, studies have shown that individuals with the highest 10% of PRS have a 3.37-fold increased risk of developing PD compared to those with the lowest decile [50]. This consistent result highlights the significant role of genetic factors in predicting PD. Our SHAP analysis underscores the importance of predictive factors in our model for PD prediction, enhancing the explainability of ML algorithms.

Our study has some limitations. Firstly, due to the limitations in the availability of PD data, our research results are solely based on the PPMI dataset, lacking external validation. Secondly, a major issue with PRS is portability across different ethnicities [51]. The predictive performance of PRS depends on GWAS, and our summary statistics are obtained from individuals mainly of European ancestry. When applied to individuals of different ethnicities, the predictive performance may be affected. Future studies are needed to evaluate our model when applied to other ethnicities, and model fine-tuning should be conducted when summary statistics from diverse ethnicities becomes available. While SHAP serves as a valuable tool for model explanation, it does have limitations [52]. Kernel SHAP uses Monte Carlo algorithms to estimate Shapley values and thus can impose a significant computational cost. Furthermore, as Kernel SHAP tries to utilize linear regression to approximate the relationship between input features and model output, it may not fully capture the complexity of the model or the non-linear relationships between features, thereby limiting a deeper understanding of the model. It is worth noting that the order of SHAP’s input could influence its results, and early involvement of domain knowledge is usually recommended and can have major benefits. Further research is needed in the field of explainable models to explore other algorithms for predicting the risk of PD. Nevertheless, our study has provided an efficient framework for PD risk prediction and shed light on the importance of each predictor.

In conclusion, our research has established a practical framework for constructing accurate risk prediction models for PD, where the penalized logistic regression model built on demographics, clinical assessments, and PRS achieved an AUC of 0.94. Our study found that UPSIT and PRS are regarded as the most important predictors for PD consistently. We believe that our study can offer a viable and promising non-invasive approach to PD risk prediction, thereby aiding in early, precise diagnosis and treatment for PD patients.

## Figures and Tables

**Figure 1 biomolecules-13-01761-f001:**
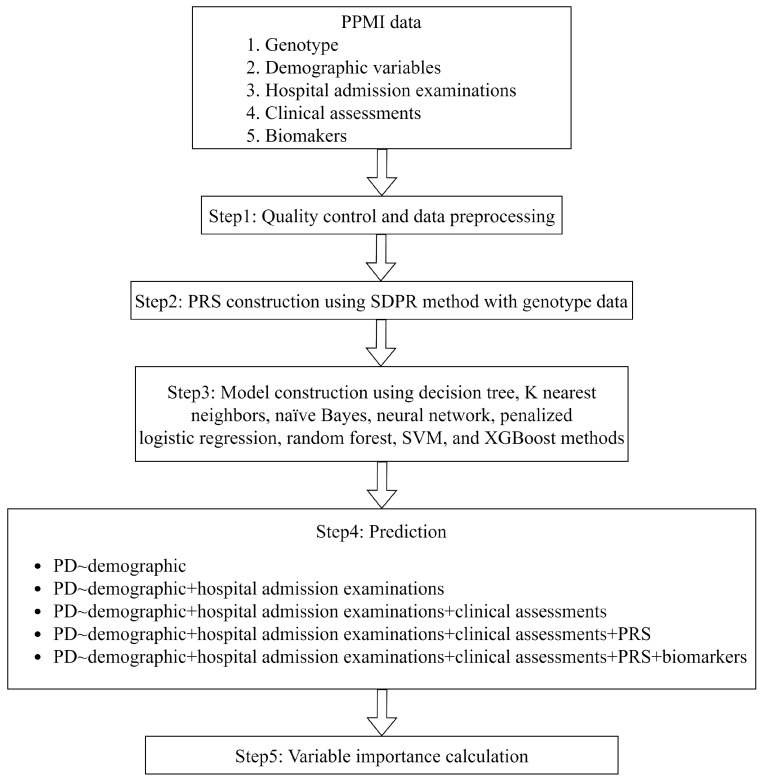
The flow chart of this study.

**Figure 2 biomolecules-13-01761-f002:**
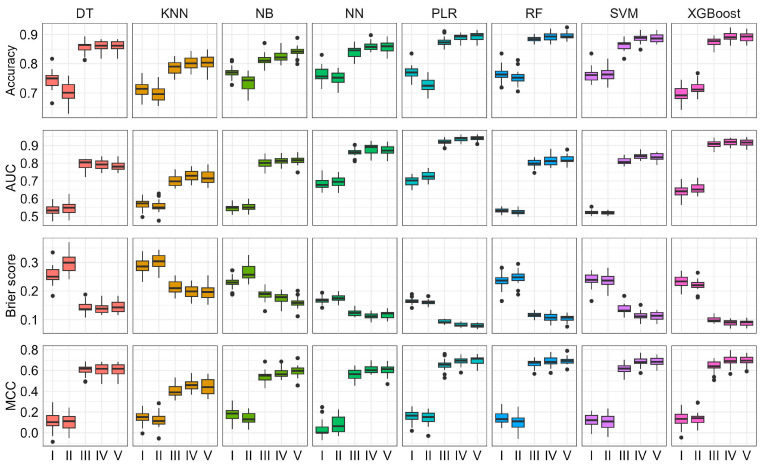
The comparisons of accuracy, area under the curve (AUC), Brier score, and Matthews correlation coefficient (MCC) among different modeling strategies. ML methods include decision tree (DT), K nearest neighbors (KNN), naïve Bayes (NB), neural network (NN), penalized logistic regression (PLR), random forest (RF), support vector machine (SVM), and Extreme Gradient Boosting (XGBoost). The scenarios assessed were as follows: Scenario I: demographic, Scenario II: demographic + hospital admission examinations, Scenario III: demographic + hospital admission examinations + clinical assessments, Scenario IV: demographic + hospital admission examinations + clinical assessments + PRS, Scenario V: demographic + hospital admission examinations + clinical assessments + PRS + biomarkers.

**Figure 3 biomolecules-13-01761-f003:**
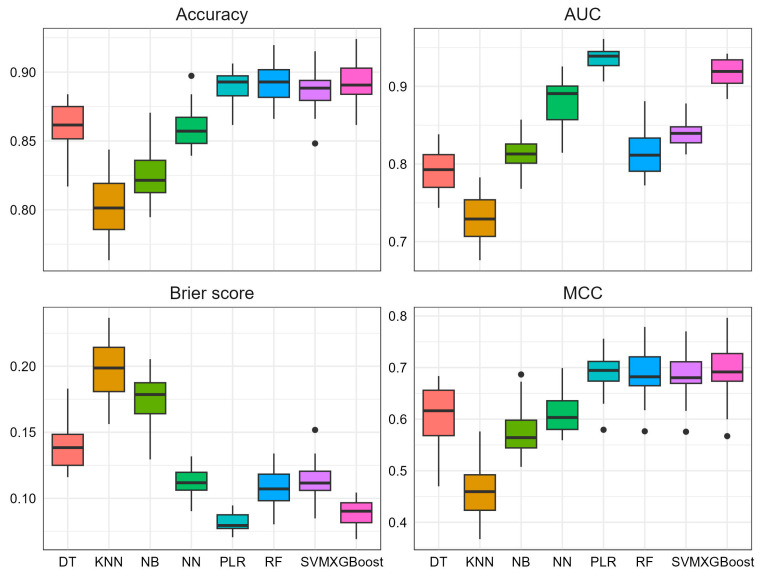
The comparisons of accuracy, area under the curve (AUC), Brier score, and Matthews correlation coefficient (MCC) among different ML methods in scenario IV (demographic + hospital admission examinations + clinical assessments + PRS). ML methods include decision tree (DT), K nearest neighbors (KNN), naïve Bayes (NB), neural network (NN), penalized logistic regression (PLR), random forest (RF), support vector machine (SVM), and Extreme Gradient Boosting (XGBoost).

**Figure 4 biomolecules-13-01761-f004:**
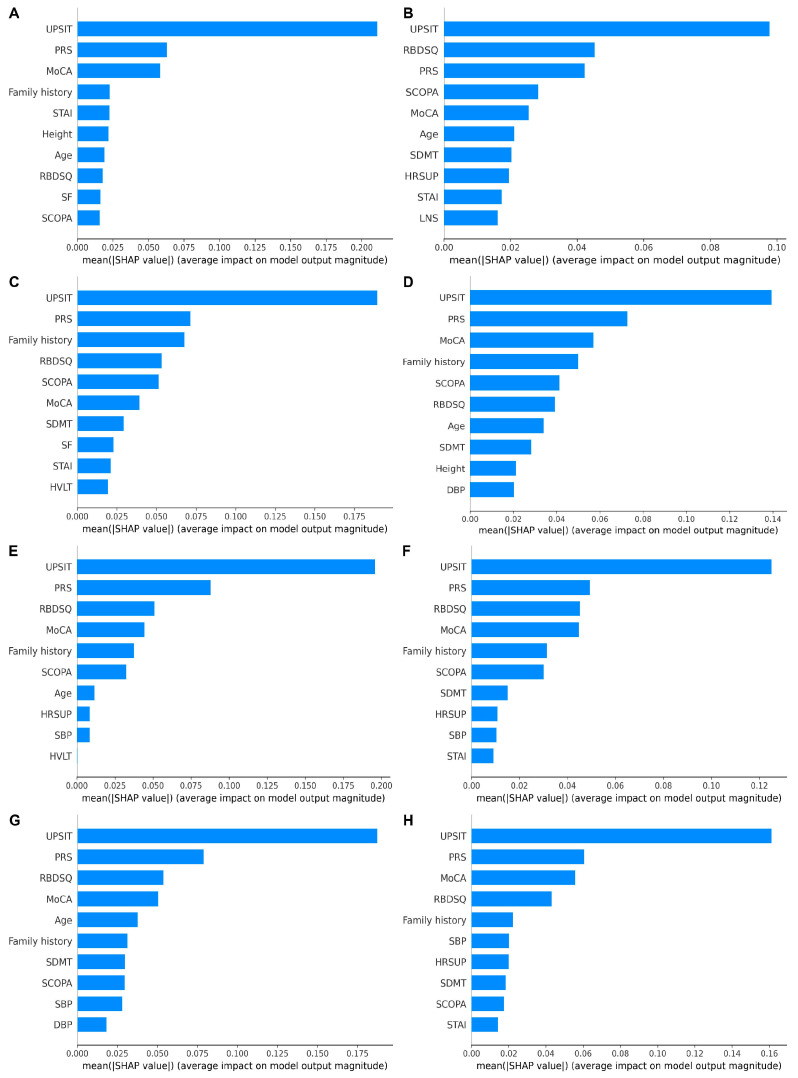
The importance ranking of the top 10 variables based on different ML methods according to the mean (|SHAP value|). The plot displays the top 10 variables in descending order based on the SHAP values, indicating their importance in the development of the final predictive model. ML methods include: (**A**) decision tree, (**B**) K nearest neighbors, (**C**) naïve Bayes, (**D**) neural network, (**E**) penalized logistic regression, (**F**) random forest, (**G**) support vector machine, (**H**) Extreme Gradient Boosting.

**Figure 5 biomolecules-13-01761-f005:**
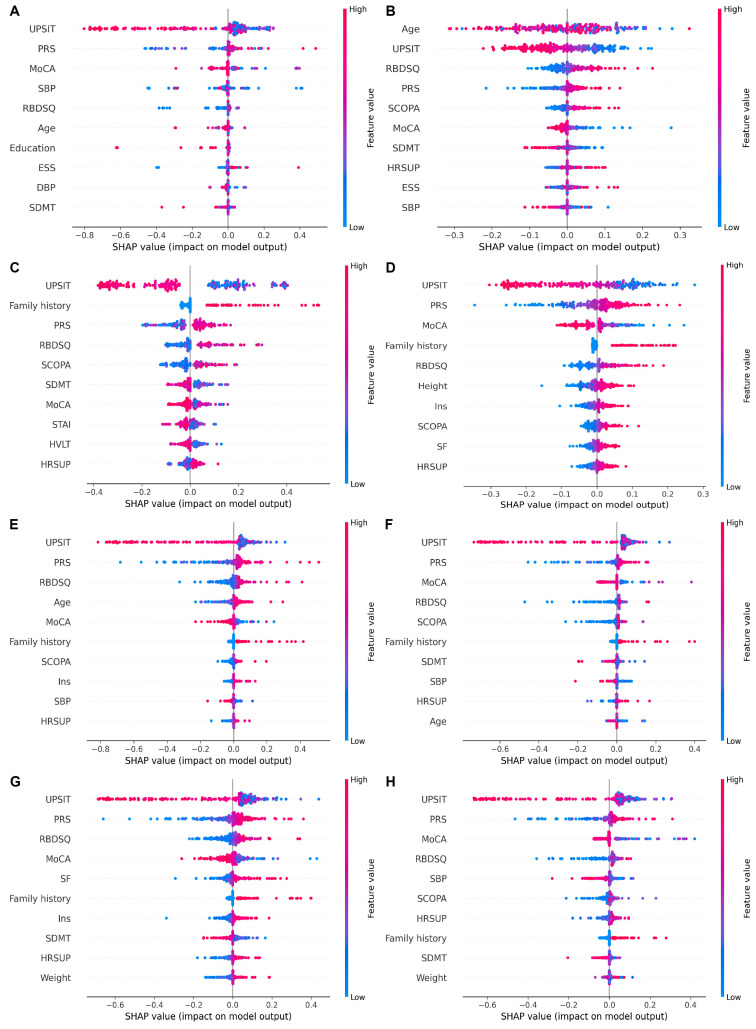
The SHAP summary plot for the top 10 important variables contributing to different ML methods in the PPMI dataset based on a random sample. ML methods include: (**A**) decision tree, (**B**) K nearest neighbors, (**C**) naïve Bayes, (**D**) neural network, (**E**) penalized logistic regression, (**F**) random forest, (**G**) support vector machine, (**H**) Extreme Gradient Boosting. Feature rankings (*y*-axis) indicate the importance to the predictive model, while the *x*-axis represents the SHAP values of the features, reflecting their positive or negative associations. The horizontal position indicates whether the impact of the value is positively or negatively associated with higher or lower predictions, where red dots represent high-risk values, and blue dots represent low-risk values.

**Table 1 biomolecules-13-01761-t001:** Summary of common clinical assessment methods and questionnaires.

Clinical Assessments	Assessment Contents
Geriatric Depression Scale (GDS)	Depressive symptoms
Questionnaire for Impulsive-Compulsive Disorders	Impulsive compulsive disorders
State Trait Anxiety Test (STAI)	State anxiety and trait anxiety level
Assessment of Autonomic Dysfunction (SCOPA)	Autonomic nervous system dysfunction
Epworth Sleepiness Scale Score (ESS)	Daytime sleepiness level
Categorical REM Sleep Behavior Disorder Questionnaire (RBDSQ)	Categorical rapid eye movement sleep behavior disorder
University of Pennsylvania Smell Inventory Test (UPSIT)	Olfactory function
The Letter Number Sequencing Test	Working memory and attention span
Semantic Fluency (SF)	Semantic memory and lexical output ability
Symbol Digit Modalities (SDMT)	Information processing speed and cognitive function
Montreal Cognitive Assessment (MoCA)	Attention, memory, executive function, and other cognitive domains
Hopkins Verbal Learning Test Immediate Recall (HVLT)	Immediate memory ability
Benton Judgment of Line Orientation	Spatial perception and direction judgment ability

**Table 2 biomolecules-13-01761-t002:** Summary statistics of demographics and selected clinical measures.

	Cases (*n* = 564)	Controls (*n* = 183)
**Demographic characteristics (S1)**		
Age, years	62.21 (9.77)	61.53 (10.75)
Sex ^a^	335 (59.40%)	118 (64.48%)
Education, years	15.38 (3.55)	16.07 (2.84)
PD family history ^a^	209 (37.06%)	9 (4.92%)
**Race** ^a^		
White	535 (94.86%)	171 (93.44%)
Black	3 (0.53%)	8 (4.37%)
Asian	8 (1.42%)	1 (0.55%)
Ethnicity	31 (5.50%)	2 (1.09%)
**Handedness** ^a^		
Left	56 (9.93%)	23 (12.57%)
Right	493 (87.41%)	147 (80.33%)
**Hospital admission examinations (S2)**		
Height	171.13 (9.83)	171.75 (10.23)
Weight	79.24 (16.30)	79.99 (15.66)
Systolic blood pressure (SBP)	130.56 (18.12)	132.78 (16.86)
Diastolic blood pressure (DBP)	77.80 (10.77)	77.74 (10.70)
Heart rate (HRSUP)	68.27 (10.89)	65.64 (9.45)
Temperature	36.49 (0.44)	36.51 (0.43)
**Neurological assessment** ^a^		
Sensory exams	54 (9.57%)	14 (7.65%)
Coordination assessments	50 (8.87%)	4 (2.19%)
Reflex evaluations	232 (41.13%)	76 (41.53%)
Motor function assessments	20 (3.55%)	1 (0.55%)
**Clinical assessments (S3)**		
Geriatric Depression Scale (GDS)	5.45 (1.60)	5.20 (1.38)
Questionnaire for Impulsive-Compulsive Disorders	0.48 (1.03)	0.30 (0.73)
State Trait Anxiety Test (STAI)	92.87 (8.75)	94.15 (6.94)
Assessment of Autonomic Dysfunction (SCOPA)	15.16 (10.58)	9.14 (7.42)
Epworth Sleepiness Scale Score (ESS)	6.05 (3.79)	5.55 (3.30)
Categorical REM Sleep Behavior Disorder Questionnaire (RBDSQ)	3.43 (2.33)	1.79 (1.78)
University of Pennsylvania Smell Inventory Test (UPSIT)	22.20 (8.60)	33.92 (4.94)
The Letter Number Sequencing Test (LNS)	10.34 (2.86)	10.85 (2.61)
Semantic Fluency (SF)	48.70 (12.34)	52.03 (11.43)
Symbol Digit Modalities (SDMT)	40.24 (10.47)	46.33 (10.32)
Montreal Cognitive Assessment (MoCA)	26.74 (2.86)	28.21 (1.11)
Hopkins Verbal Learning Test Immediate Recall (HVLT)	24.27 (5.11)	26.03 (4.49)
Benton Judgment of Line Orientation	12.42 (2.52)	13.05 (2.02)
**Biological variables (S4)**		
Alpha-synuclein (α-Syn)	1553.98 (595.64)	1748.63 (719.36)
Amyloid β1-42 (Aβ_42_)	921.29 (359.77)	1053.66 (481.70)
Total tau (t-tau)	173.15 (56.95)	196.99 (76.47)
Phosphorylated tau (p-tau)	14.66 (5.27)	17.41 (8.23)
Total serum uric acid	312.43 (76.46)	324.36 (77.98)

^a^ For sex, PD family history, race, handedness, and neurological assessment, the number and its percentage are reported. For all others, mean (standard deviation) is reported.

**Table 3 biomolecules-13-01761-t003:** The average predictive performance of each machine learning model based on 20 random samples under different modeling strategies.

Algorithms	TPR	TNR	MCC	Brier Score	Accuracy	AUC
**Scenario I**: Demographic variables						
DT	93.616%	13.052%	0.102	0.255	0.745	0.534
SVM	98.155%	6.103%	0.118	0.237	0.763	0.523
NB	97.101%	11.925%	**0.177**	0.231	**0.769**	0.546
KNN	84.187%	29.484%	0.149	0.288	0.712	0.570
RF	97.452%	9.014%	0.143	0.236	0.764	0.533
PLR	98.477%	7.887%	0.156	**0.165**	**0.769**	**0.698**
NN	99.561%	1.690%	0.034	0.167	0.763	0.684
XGBoost	81.933%	30.141%	0.126	0.233	0.696	0.642
**Scenario II**: Demographic variables + Hospital admission examinations						
DT	83.748%	25.540%	0.106	0.301	0.699	0.549
SVM	98.653%	4.883%	0.105	0.236	**0.764**	0.521
NB	89.898%	20.751%	0.140	0.265	0.735	0.554
KNN	82.489%	28.920%	0.123	0.302	0.698	0.558
RF	95.959%	9.107%	0.098	0.247	0.753	0.525
PLR	98.184%	7.418%	**0.145**	**0.161**	0.727	**0.728**
NN	96.310%	7.512%	0.081	0.176	0.752	0.696
XGBoost	86.296%	25.352%	0.133	0.220	0.718	0.660
**Scenario III**: Demographic variables + Hospital admission examinations + Clinical assessments						
DT	92.123%	65.915%	0.602	0.141	0.859	0.791
SVM	91.127%	70.423%	0.619	0.138	0.862	0.811
NB	82.460%	77.183%	0.546	0.188	0.812	0.799
KNN	86.120%	54.366%	0.407	0.214	0.786	0.704
RF	95.520%	65.634%	**0.662**	0.116	**0.884**	0.799
PLR	92.123%	72.582%	0.652	**0.092**	0.875	**0.920**
NN	90.395%	64.131%	0.560	0.125	0.842	0.862
XGBoost	93.148%	69.014%	0.640	0.100	0.874	0.906
**Scenario IV**: Demographic variables + Hospital admission examinations + Clinical assessments + PRS						
DT	92.533%	65.446%	0.605	0.139	0.861	0.790
SVM	93.324%	73.615%	0.682	0.114	0.886	0.839
NB	83.660%	78.967%	0.575	0.175	0.825	0.813
KNN	87.145%	58.779%	0.460	0.196	0.804	0.731
RF	96.076%	67.042%	0.686	0.108	**0.892**	0.812
PLR	93.704%	73.333%	0.687	**0.082**	0.889	**0.937**
NN	91.362%	68.920%	0.610	0.112	0.860	0.881
XGBoost	94.348%	72.864%	**0.693**	0.089	0.892	0.919
**Scenario V**: Demographic variables + Hospital admission examinations + Clinical assessments + PRS + Biological variables						
DT	92.006%	64.789%	0.675	0.145	0.855	0.810
SVM	93.763%	72.864%	0.717	0.112	0.888	0.833
NB	86.413%	76.526%	0.640	0.159	0.841	0.830
KNN	87.906%	55.587%	0.477	0.198	0.802	0.718
RF	96.310%	67.700%	0.740	0.105	**0.895**	0.820
PLR	94.085%	73.897%	0.719	**0.079**	0.893	**0.939**
NN	90.970%	68.809%	0.547	0.117	0.857	0.873
XGBoost	94.348%	72.864%	**0.768**	0.088	0.892	0.914

Decision tree (DT), K nearest neighbors (KNN), naïve Bayes (NB), neural network (NN), penalized logistic regression (PLR), random forest (RF), support vector machine (SVM), and Extreme Gradient Boosting (XGBoost); true positive rate (TPR), true negative rate (TNR), Matthews correlation coefficient (MCC), area under the curve (AUC).

## Data Availability

The data analyzed in this study is subject to the following licenses/restrictions: The datasets can be found at: http://www.ppmi-info.org/ (accessed on 20 August 2023) for PPMI, and they can be requested from PPMI studies. Summary statistics are available from: https://drive.google.com/file/d/1FZ9UL99LAqyWnyNBxxlx6qOUlfAnublN/view, accessed on 20 August 2023.

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
