# Peer review of "Prediction of Parkinson’s Disease Using Machine Learning Methods"

_biomolecules, 2023, doi:10.3390/biom13121761_

Round 1

Reviewer 1 Report

Comments and Suggestions for Authors

Author Response

We have provided a point-by-point response to the comments from the reviewers. Please see the attachment.

Reviewer 2 Report

Comments and Suggestions for Authors

Authors compare different machine learning algorithms to predict Parkinson's disease.

Overall, in the following, the strengths and the weaknesses of the paper are summarized.

Strengths:

 (+) The paper is well-written.

 (+) The proposed method is well-explained.

 (+) The experiments are convincing.

 (+) The figures are appropriate.

Weaknesses:

 (-) Novelty is unclear.

 (-) There are English issues.

 (-) References are inadequate.

 (-) The introduction must be improved.

 (-) The related work section must be enhanced.

 (-) Some improvements are needed in the description of the method.

In the following more detailed comments about the manuscript:

==== ENGLISH ==== 

There should always be a single space before characters such as ( and [. 

==== INTRODUCTION ==== 

The introduction should clearly explain the key limitations of prior work that are relevant to this paper.

Contributions should be highlighted more. It should be made clear what is novel and how it addresses the limitations of prior work. 

Explainable AI should be mentioned in the abstract and the Introduction paragraph. Also, Shap methodology should be better explained as well as to interpret Shap graphs.

==== RELATED WORK ==== 

The related work section is not well organized. Authors must try to categorize the papers and present them in a logical way.

The authors should explain clearly what the differences are between the prior work and the solution presented in this paper.

Particularly, you should deeply investigate the literature on machine learning for PD, with a particular interest in explainable models, since you apply Shap to explain your results. 

Some examples are: https://doi.org/10.1109/ICDABI51230.2020.9325709 where authors applied a variable adaptive moment-based backpropagation algorithm of ANN called BPVAM and they studied the influence of dimensionality reduction on the classification results, https://doi.org/10.1007/978-3-031-39059-3_22 propose an explainable Deep Learning approach for the detection of PD from single photon emission computed tomography (SPECT) images, https://doi.org/10.1038/s41598-023-42542-y a recent work published on Scientific Report on the use of Shap values for medical purposes, etc.

==== METHOD ==== 

A novel solution is presented but it is important to explain better the design decisions (e.g. why the solution is designed like that)

Particularly, it seems you propose a multimodal analysis of PD disease. You should describe how data with different dimensions have been combined. Also, figure 1 mentions Genotype data that disappear in the following box. On the contrary, you mention PRS data, without describing them.

The experimental setting must be clearly described. Training and test set percentage. Random/Stratified. Target classes. Did you use cross-validation? If not, you should. If yes, please include it in the paper.

Figure 2 shows boxplots for different measures, methods, and scenarios. It is not clear how did you obtain these boxplots. Why do you have different values for a given measure, method, and scenario? This has not been described, and it must be clarified

Which hyperparameters have you used for each ML method? A detailed description of the hyperparameters and the methodology to identify them (e.g. hyperparameter tuning/standard parameters) should be added.

Table 3 shows measures for different methods and scenarios. Are these values obtained in a training-test setting? In a cross-validation setting? If this is the case, are they the average values over different folds? These details must be included in the paper, to make experiments reproducible.

Shap graphs show the first 10 features. My doubt is related to the aforementioned multimodal data. Which data have you analyzed with the shap graphs? Is the dataset obtained joining different types of data? did you focus on a single kind of data? This must be clarified. For example, I cannot find UPSIT feature, that is used in the Shap graphs, in any scenario in Table 2. Features used in data must be clearly described, and there must be traceability among all the sections of the paper.

On page 15 you discuss the concept of interpretability and you mention some references. Theoretical discussion must be placed in the first part of the manuscript, then the experiments must be designed, and then the results must be discussed. I suggest sticking to this organization since it makes the manuscript more clear. 

I would not use the term interpretability, indeed Shap is an explainable method for agnostic explanations. I recommend briefly introducing the concept of Explainable AI, adding related, recent, references, such as https://doi.org/10.1016/j.inffus.2019.12.012 Explainable Artificial Intelligence (XAI): Concepts, taxonomies, opportunities and challenges toward responsible AI

Finally, recent work has shown the limits of visual explanations, such as the Shap graphs, and the effectiveness of natural language-based explanations to support better communication of the information embedded in the Shap values. Please refer to https://doi.org/10.1016/j.ins.2022.10.010 for more details.

Author Response

(The authors gave the same response as above.)

Round 2

Reviewer 2 Report

Comments and Suggestions for Authors

Authors have properly enriched their work, by addressing each comment in a suitable way. The paper turns out to be notably improved.